# Unusual Suspects: Bone and Cartilage ECM Proteins as Carcinoma Facilitators

**DOI:** 10.3390/cancers15030791

**Published:** 2023-01-27

**Authors:** Alexandra Sorvina, Michael Antoniou, Zahra Esmaeili, Marina Kochetkova

**Affiliations:** Centre for Cancer Biology, An Alliance Between SA Pathology and the University of South Australia, Adelaide, SA 5000, Australia

**Keywords:** bone, cartilage, extra-cellular matrix (ECM), ectopic expression, cancer

## Abstract

**Simple Summary:**

To provide insights into the role of the extracellular matrix (ECM) in health and pathological conditions, it is important to identify tissue-specific proteins, their interacting networks and functions. Latest discoveries suggest that multiple tumors express, and use to their advantage, atypical ECM components that are not found in the cancer tissue of origin. The aim of this review was to summarize and critically assess available information on the expression and function of atypical carcinoma-, bone- and cartilage-specific components of the extracellular matrix. To the best of our knowledge, this topic has not previously been covered by any published review, and thus provides a novel perspective for devising strategies to target tumor stroma as anti-cancer therapeutic options.

**Abstract:**

The extracellular matrix (ECM) is the complex three-dimensional network of fibrous proteins and proteoglycans that constitutes an essential part of every tissue to provide support for normal tissue homeostasis. Tissue specificity of the ECM in its topology and structure supports unique biochemical and mechanical properties of each organ. Cancers, like normal tissues, require the ECM to maintain multiple processes governing tumor development, progression and spread. A large body of experimental and clinical evidence has now accumulated to demonstrate essential roles of numerous ECM components in all cancer types. Latest findings also suggest that multiple tumor types express, and use to their advantage, atypical ECM components that are not found in the cancer tissue of origin. However, the understanding of cancer-specific expression patterns of these ECM proteins and their exact roles in selected tumor types is still sketchy. In this review, we summarize the latest data on the aberrant expression of bone and cartilage ECM proteins in epithelial cancers and their specific functions in the pathogenesis of carcinomas and discuss future directions in exploring the utility of this selective group of ECM components as future drug targets.

## 1. Introduction

The identification of extracellular matrix (ECM) proteins and their underpinning tissue-specific networks is a key step towards providing insights into the role of the ECM in health and pathological conditions. The ECM is a three-dimensional supramolecular entity, with a unique composition and topology, that provides architectural/mechanical support and biochemical cues for tissue morphogenesis, differentiation and homeostasis. Although the ECM composition can be broadly described as a combination of water, proteins and polysaccharides (the proportions of which are defined by the functional requirements of the tissue [1]), the importance of the ECM is clearly demonstrated by its role in tumorigenesis and metastasis [2,3,4]. The ECM provides a dynamic and reciprocal dialogue between cells (e.g., epithelial, fibroblast, adipose and endothelial cells) and various matrix components; cells can remodel the ECM, whilst the matrix can influence cell behavior (“dynamic reciprocity” [5]). The loss of correct matrix organization, crosstalk and homeostasis is a hallmark of solid tumors, with both non-malignant and malignant cells contributing to matrix deposition and remodeling.

In cancer, remodeling of the interstitial ECM results in matrix stiffness, biophysical/biochemical changes that alter signaling pathways, cell migration and tumor progression. Indeed, tissue fibrosis, inflammation (which potentiates stromal fibroblast activation) and cancer are inextricably connected [6], with desmoplasia being commonly observed in several solid tumors. A correlation between desmoplasia and cancer is well recognized in breast cancer [7], pancreatic cancer [8] and metastatic lymph nodes [9]. In fact, establishment of this tumor-supportive microenvironment is required for the maintenance of multiple processes that govern tumor development, progression and metastasis, which includes: changes in the abundance and composition of ECM components; modifications at the post-translational level; proteolytic degradation, followed by formation of matrikines; and architectural remodeling of ECM for opening cell migratory passages [2]. A large body of experimental and clinical evidence has shown that altered expression of matrices in tumors is associated with cancer onset and progression and reduced response to treatment due to altered apoptosis [10]. The latest evidence also suggests that multiple cancer types express, and use to their advantage, atypical ECM components that are not found in the cancer tissue of origin.

The exact composition and organization of the ECM vary between organs and dictate its distinct functions. To allow tissue homeostasis and repair and adaptation to a changing environment, the ECM is constantly renewed by fibroblasts synthesizing ECM components, while also secreting ECM-degrading enzymes. Cancers, like normal tissues, actively modify their ECM to serve their specific growth requirements. Additionally, the extreme plasticity of tumor cells leads to the activation of corrupt transcription programs, resulting in the production of atypical ECM components that are not found in the cancer tissue of origin, but which are important for cancer pathogenesis and, thus, may represent specific targets for cancer diagnosis and treatment.

Cancer that arises in the epithelia is the most common type and is found in most organs of the human body. In this review, we summarize cancer-specific expression patterns of bone and cartilage ECM proteins and their roles in specific tumor types as an overview of these ectopically expressed ECM proteins in cancer is missing. We also discuss future directions, exploring the utility of the identified ECM components as diagnostic and prognostic markers, as well as novel therapeutic targets for cancer treatment. Outside the scope of this review are the expression and function of atypical ECM components in non-epithelial cancers that have also been documented.

## 2. Cancer-Related Functions of the ECM

The ECM is a complex meshwork of highly crossed-linked proteins that provides architectural support, biochemical cues and anchorage for the cells that comprise the tissue. There are two basic types of ECM, which are distinguished by their location and composition [11], including the interstitial matrix and pericellular matrix (a layer that surrounds cells; for instance, the basement membrane). The interstitial matrix constitutes the bulk of the ECM, while the basement membrane interacts directly with the epithelium and endothelium to which epithelial cells can anchor, providing cues for the establishment of apico-basal polarity and cell differentiation. Although ECM can be deposited by any cell type, the major producers are fibroblasts and myofibroblasts, which are the primary mediators of fibrosis [12]. The potent stimulator of fibrosis is the transforming growth factor beta (TGF-β1) that can enhance expression of the ECM remodeling gene *ACTA2* (alpha-smooth muscle actin (α-SMA)) [13], *PLOD2*, encoding procollagen-lysine, 2-oxoglutarate 5-dioxygenase 2 that is required for collagen deposition [14,15] and numerous other pro-fibrotic proteins. The secretion of both pro-fibrotic and inflammatory factors, such as transforming growth factor alpha, TGF-β1, fibroblast growth factor 2 (FGF2), platelet-derived growth factor and epidermal growth factor, is orchestrated by cancer cells, and they can induce differentiation of stromal cells into cancer-associated fibroblasts (CAFs) to support tumor growth [16]. CAFs can exhibit heterogeneity and are believed to have multiple origins (e.g., tissue-resident or bone-marrow-derived fibroblasts and adipocytes [17,18]), and produce large amounts of structural ECM proteins, including fibrillar and non-fibrillar collagens, elastic fibers, glycoproteins and proteoglycans. These proteins provide a supportive framework, with laminin and fibronectin forming bridges between these structural components to strengthen it and to connect cells with the ECM and increase matrix stiffness around the tumor [19].

The increased deposition of collagenous proteins and changes to fiber alignment leads to the alteration of ECM homeostasis, representing key characteristics of cancer that are associated with poor survival rates [20,21,22,23]. The perturbation of ECM homeostasis affects the biochemical and biophysical properties of the matrix, both of which are crucial for normal tissue function. For example, in breast cancer, three tumor-associated collagen signatures have been identified: densely packed collagen around the tumor, spheroidal shells surrounding deposited collagen, and linear outgrowths of collagen leading into the breast parenchyma [24,25]. It has been shown that breast cancer becomes more invasive upon increased collagen bundling, with invasion of ductal breast cancer cells into collagen matrices mediated by lysyl oxidase-like 3 that reinforces local matrix stiffness [22]. Moreover, ECM fiber alignment can enhance cell-ECM interactions and facilitate the migration of cancer cells [26]. Once synthesized, the ECM proteins undergo post-translational modifications that affect matrix interactions with other molecules and cancer cell motility [27]. Concentration gradients for haptotactic migration or pattern formation can be created by binding of the ECM proteins to soluble/secreted factors and growth factors [28]. This interaction is bidirectional [29] and is critical for cell–ECM interactions.

Cancer cell–ECM interaction can activate several pathways related to mechanotransduction, mediated by integrins, discoidin domain receptors and syndecans [30,31]. For example, sensing and adaptation to tissue stiffness occurs by binding breast myoepithelial cells to fibronectin, either through constitutively expressed integrin α_5_β_1_ in healthy tissue, or selectively expressed α_v_β_6_ in cancer [32]. Indeed, integrins have a close relationship to force, conveying mechanical stresses bidirectionally across the plasma membrane and transducing mechanical forces into chemical signals [33]. The cell–ECM contact guides cell polarization, including intracellular location of the nucleus and the microtubule organizing center, and this is controlled by lamin A/C that acts as a mediator in ECM sensing and signal transduction [34]. Transmission of the mechanical stress (e.g., osmotic compression) to the cells is mediated by the ECM and this impacts cell proliferation and migration in a 3D environment [35]. Moreover, the ECM aids in the spatial organization of intercellular junctions; the magnitude of intra-/intercellular forces, controls cell–cell interactions [36] and serves as an adhesive substrate for cell migration.

The ECM components undergo proteolytic and non-proteolytic degradation as part of the remodeling process. This includes replacement of normal ECM with tumor-derived ECM. Multiple target-specific proteases [37] support cancer-cell motility by opening migratory tracks and reducing the mechanical stress on migrating cells. For instance, the overexpression of matrix metallopeptidase 11 (MMP-11) in macrophages increases migration of human epidermal growth factor receptor 2 (HER2)-positive breast cancer cells by activating the MAPK pathway via binding of the chemokine CCL2 to its receptor on cancer cells, followed by activation of the MAPK pathway and upregulation of MMP-9 [38]. Single cell invasion can be suppressed by the dense matrix, while collective cell migration is less sensitive to its density, with MMP secretion increasing the rates of cell migration independent of cell–cell adhesion [39]. In addition, there are several types of cellular proteolytic surface structures rich in filamentous actin, β1 integrin and MT1-MMP that can cleave and realign ECM to facilitate migration [40]. Growth factors can be tethered to the ECM, protecting them from degradation, and be activated by proteases allowing their spatial-temporal availability required for tumor progression. Moreover, ECM molecules can be cleaved by proteases, producing a range of bioactive fragments, which regulate numerous biological processes [41] and can be shed into the circulation [42]. These fragments, together with other ECM molecules, can serve as diagnostic and prognostic markers in different epithelial cancers.

It is important to note, however, that due to limited availability of a robust methodology for the assessment of the precise ECM composition of both normal tissues and cancers, knowledge of the ECM landscape and the specific functions of its individual members is still sparse.

## 3. Matrisome Components/Classes

The matrisome or global composition of the ECM is composed of ~300 unique matrix macromolecules, which can be classified into core matrisome and matrisome-associated proteins [43,44]. Core ECM proteins represent ~1–1.5% of the mammalian proteome, excluding alternatively spliced isoforms, and comprise glycoproteins, collagens and proteoglycans [45]. The matrisome-associated proteins include ECM-affiliated proteins (i.e., architecturally similar to the ECM proteins or those that associate with them), ECM regulators (e.g., remodeling enzymes, proteases and crosslinkers) and secreted factors [45]. Although the ECM of different source tissues shares common proteins, there is a subset of matrisome proteins that display tissue-specific expression (also known as the ‘matrisome signature’) [45], which can be remodeled upon metastasis and chemotherapy [46]. The proteins are secreted both by cancer cells and stromal cells. These tumor- and stroma-derived ECM molecules also differ in cancers with different metastatic potential [47].

The proteolytic processing of some ECM components leads to the production of bioactive fragments, such as matrikines or matricryptins, that can mediate cancer-cell invasiveness. There are two major classes of matrikines; natural matrikines and cryptic matrikines that are inactive in the mature/secreted form until undergoing conformational changes [48]. The major sources of matrikines are collagens, glycoproteins, elastin and laminins [49].

Bone and cartilage are two specialized forms of connective tissues, which are composed of cells embedded within a matrix. Cartilage, bone, tendon and ligaments are all composed of an ECM made of collagens, proteoglycans and specialized glycoproteins that are actively synthesized, precisely assembled and subsequently degraded by the resident connective tissue cells.

### 3.1. Bone-Specific Matrisome

The two main types of bone composition are cortical and trabecular bone. These two types have differing structural properties, affecting the way in which each responds to mechanical loading [50]. Constant remodeling of bones is carried out throughout life to maintain robust structure and function. Differing from that of other connective tissue, bone is separated by a layer of osteoblasts that are connected by tight and gap junctions. These osteoblasts mainly produce lamellar type I collagen that is extremely dense and heavily cross-linked by several enzymes [51,52].

While ~90% of the bone ECM is composed of type I collagen, there are still quantitatively ~5% of the ECM protein components that present interesting properties [51]. Proteomic analysis of decalcified bone suggests that there are well over 100 ECM proteins that make up these bone-specific ECM components [53,54]. It is the components of this unique bone ECM that can induce the production of new osteoblast-lineage cells, such as mesenchymal stem cells, osteocytes, and osteoclasts. The non-collagenous bone proteins can be divided into four groups: γ-carboxyglutamate-containing proteins, proteoglycans, glycoproteins and small integrin-binding ligands, N-linked glycoproteins (SIBLINs) [55]. The bone ECM also contains inorganic components of mainly calcium-deficient apatite and trace elements [56].

### 3.2. Cartilage-Specific Matrisome

Another unique and multifaceted tissue is the cartilage, specially adapted to bear compressive loads and significant shear force [57]. In the human body the cartilage is divided into elastic, fibrocartilage, fibroelastic or hyaline cartilage depending on the composition of the matrix [58]. The protein structure of these types of cartilage is primarily composed of collagen type II, combined with an interlocking mesh of fibrous proteins and proteoglycans, hyaluronic acid and chondroitin sulfate [59]. The cartilage space is subject to a very harsh biochemical environment and is also devoid of blood vessels, lymphatics, and nerves. Much of the cartilage matrix is then composed of tissue fluid, accounting for ~70–80% of the total cartilage weight [60]. These large, hydrated spaces of ECM are created by aggrecan, the most abundant proteoglycan within cartilage [61].

Formation of the cartilage arises from the mesenchyme, with some of these cells aggregating to form a blastema. Within this blastema, the secretion of the cartilage matrix begins; the cells are now referred to as chondroblasts. A specialized and tough matrix is formed and the cells encased within are then called chondrocytes [62]. This chondrocyte cell type is responsible for the development, maintenance and repair of the cartilage ECM [60]. The chondrocytes are responsible for fluid exchange within the matrix as well and account for ~1–6% of the total ECM mass within the cartilage [63]. While self-repair in cartilage is uncommon, the ECM is continuously remodeled [59,64], producing mechanical, electrical and chemical signaling that can have an effect on the key functions of chondrocytes [62].

Ectopic expression of these unique bone and cartilage matrix proteins [56] has been reported in non-skeletal tissues as well as in epithelial tumors (Figure 1, detailed summary in Table 1). Their presence in solid tumors, at sites where they present no homeostatic function, poses an intriguing question as to their regulation and function in carcinomas.

## 4. Bone- and Cartilage-Specific ECM Components in Cancer Progression, Spread and Invasion

### 4.1. Collagenous Proteins

The ectopic expression of bone [56] and cartilage [65] collagenous proteins has been reported in various epithelial tumors. Collagens are characterized into 28 types, with all containing specific amino acid sequences encoding one or more triple-helical domains [66]. The ECM of bone and cartilage is composed predominantly of a fibril-forming collagen [67], which provides structural and functional integrity of the tissue under physiological conditions, and ‘minor collagens’, which are present at low quantities but have vital functions. However, deposition of a fibrotic, collagen-rich ECM within TME can result in cancer establishment and progression and failure in the therapeutic response [68].

Collagen type II is the most abundant fibril-forming collagen of hyaline, fibrocartilage and elastic cartilages, representing ~50% of the protein and 80–95% of the total collagen content [69]. Collagen type IIA (containing an additional exon 2 in the NH_2_-propeptide that encodes a von Willebrand factor C domain) and IIB are produced by alternative splicing of exon 2 in the pre-mRNA and are spatially expressed during development and chondrogenesis, with *collagen type IIA* mRNA expressed in chondroprogenitor cells and *collagen type IIB* expressed by chondrocytes [69]. Like other fibril-forming collagens (i.e., types I, III, V, XI, XXIV and XXVII), collagen type IIB is synthesized as a procollagen form that is cleaved by ADAMTS-3 (ADAM metallopeptidase with thrombospondin type 1 motif 3) to remove the NH_2_-terminal propeptide domain prior to its incorporation into fibrils in the matrix [70,71]. The NH_2_-propeptide of collagen type IIB can induce death of chondrosarcoma (hCh-1), cervical cancer (HeLa) and breast cancer (MDA-MB231) cell lines via interaction between the RGD sequence present in the propeptide and the cellular integrins α_V_β_3_ and α_V_β_5_ [72]. Unlike collagen type IIB, collagen type IIA is deposited into the ECM as pN-procollagen, without removal of the NH_2_-propeptide, which is used for binding of growth factors, such as bone morphogenetic protein (BMP-2) and TGF-β1 through the von Willebrand factor C domain [73]. Collagen type II is synthesized and secreted by stromal fibroblasts, forming a collagen-rich ECM that supports endothelial cell migration and angiogenesis in high-grade serous carcinoma, as a response to increased levels of the initiator methionine tRNA (*tRNA_i_^Met^*) [74].

A minor fibril-forming collagen type V is one of the predominantly secreted collagens in the bone matrix and is essential for the fibrillation of collagen types I and III. *COL5A1* has been shown to be upregulated in multiple epithelial cancers [75], including lung adenocarcinoma metastasis [76] and invasive ductal carcinoma with distant metastasis [77], while *COL5A2* is overexpressed in bladder cancer [78]. Colorectal tumors co-express *COL5A2* and *COL11A1* [79], with COL5A2 protein binding to DDR1 and upregulating the WNT/β-catenin and PI3K/mTOR signaling pathways [80]. Co-expression of the high (COL5A1)2/low (COL5A2) heterotrimer has been identified as an unfavorable prognostic factor for patients with tongue squamous cell carcinoma [81]. A decrease in collagen type V is linked to a decrease in tumoral and microvessel apoptosis, as well as reduction in collagen types I and III in non-small-cell lung cancer [82]. Ablation of α3(V) in MMTV-PyMT mice slows tumor growth within mammary glands [83]; this agrees with previous findings showing that this protein is overexpressed in breast cancer invasive ductal carcinoma and is regulated by TGF-β1 [77]. Production of collagen type V increases during the progression of pancreatic ductal adenocarcinoma, and this is mediated by pancreatic stellate cells in the stroma [84], promoting cell migration and actin polymerization via the COL5A1-PPRC1-ESM1 axis, as has been shown in glioblastoma [85].

Collagen type IX is an integral cartilage ECM protein that belongs to the FACIT collagen subfamily and is present in the chondrocytes of growth plate cartilage, adult articular cartilage and intervertebral discs. mRNA, encoding collagen type IX, is alternatively spliced, generating isoforms that display a temporospatial expression pattern during mouse embryonic development [86]. Genetic variants in the *COL9A1* gene (i.e., *rs550675*) have shown an association with oral squamous cell carcinoma in the male population [87]. Epithelial–mesenchymal transition (EMT), invasion and migration in gastric cancer is mediated by ubiquitin-specific protease 3, which interacts with and deubiquitinates/stabilizes COL9A3 and COL6A5 [88]. High expression levels of both ubiquitin-specific protease 3 and COL9A3, compared to those of only one or low expression levels, result in the worst outcomes for patients with gastric cancer [88].

Collagen type X is a homotrimeric protein that constitutes ~1% of total collagen in adults and is produced by hypertrophic chondrocytes, with exclusive localization in the hypertrophic cartilage and the calcified zone of articular cartilage [89,90,91]. Bioinformatics analysis has demonstrated that *COL10A1* is upregulated in breast cancer, with expression levels positively correlated with estrogen receptor, progesterone receptor, HER2 and nodule status [92]. Collagen type X localizes to tumor vasculature [93] and directly interacts with the prolyl 4-hydroxylase beta polypeptide in breast cancer [94], promoting malignant progression. Similarly, high levels of *COL10A1* have been observed in gastric cancer, representing a key independent predictor of poor outcomes [95], probably because of COL10A1-mediated upregulation of lymphoid enhancer-binding factor 1 and Wnt2 [96]. Immune infiltration in pancreatic cancer correlates with high *COL10A1* expression levels, with a new *TUG1/miR-144-3p/COL10A1* axis identified upstream of the non-coding RNA pathway [97]. The progression and invasion of colorectal cancer can be promoted by visinin-like 1 protein via targeting of COL10A1 [98].

Collagen type XII is a homodimer found in association with type I collagen-containing fibrils, with a known amplification of the *COL12A1* gene and overexpression in colorectal cancer [99,100]. This collagen is expressed by osteoblasts and localizes to areas of bone formation and tendon. There are two splice variants of collagen type XII, ‘large’ XIIA and ‘small” XIIB, that have differential expression [101]; however, their functional significance is yet to be discovered. Collagen type XII is secreted by CAFs and regulates the organization of collagen type I, creating a pro-invasive environment for cancer dissemination [102]. Enrichment analysis of genes in the *COL12A1* neighborhood in colorectal cancer identified focal adhesion and PI3K-AKT signaling pathways [99]. A positive correlation was found between elevated levels of *COL12A1* and tumor invasiveness, metastasis, and advanced clinical representation of gastric cancer [103].

Collagen type XXIV is a fibril-forming collagen, expressed by differentiating osteoblasts in the forming skeleton and has a possible role in the formation of a mineralization-competent bone matrix [104]. There are two spliced products that differ in 5′-untranslated sequences [105]. The mRNA expression of *COL24A1* is significantly upregulated and associated with tumor size in patients with squamous cell carcinoma of the head and neck [106]. Similarly, *COL24A1* is upregulated in hepatocellular carcinoma compared to normal hepatic tissue and is indicative of poor prognosis for cancer patients [107].

### 4.2. Proteoglycans

The small leucine-rich proteoglycans (SLRP; 18 members categorized into five classes) are major non-collagen components of the ECM that bind to various extracellular receptors/ligands (e.g., TGF-β1, collagens, fibronectins, etc.) through their bare β-sheets that are present on the concave surface of leucine-rich receptors and regulate cell-matrix homeostasis [108]. These proteins are functionally involved in bone development and homeostasis, including cell proliferation, formation of connective tissue, organic matrix deposition, mineral deposition and bone remodeling [109,110]. Although biglycan, asporin and decorin (key constituents of the bone ECM) belong to the class I SLRP, they possess both pro- and anti-tumorigenic potential and have different roles in the pathogenesis of different cancers [111]. Upregulation of biglycan (BGN; also known as proteoglycan-1 and dermatan sulfate PG-1) has been reported in multiple types of solid tumors, including prostate [112], pancreatic [113], gastric [114], colon [115], endometrial [116,117,118], bladder [119] and breast cancers [120]. For instance, in prostate cancer, the expression of biglycan is linked to *TMPRRS2:ERG* fusion, *PTEN* deletion and androgen receptor levels [112]. In pancreatic cancer, biglycan is overexpressed, leading to the inhibition of TGF-β1-responsive and -unresponsive cancer cells through induction of G1-arrest, associated with an increase in p27 and reduction in cyclin A and proliferating cell nuclear antigen [113]. While invasion and metastasis of gastric cancer is significantly increased upon upregulation of biglycan that activates the focal adhesion kinase (FAK) signaling pathway (inducing phosphorylation of FAK and Paxillin) [114], tumor angiogenesis has been linked to biglycan-mediated regulation of vascular endothelial growth factor (VEGF), correlated with activation of the ERK signaling pathway in colon cancer [115]. In addition to sequestering in the ECM, biglycan can exist as a soluble molecule upon cleavage from the ECM (proteolytic digestion or secreted by activated macrophages) that acts as a damage-associated molecular pattern (DAMP) protein [121]. Biglycan is a ligand for macrophage toll-like receptor 2 (TLR2) and TLR4, with induced signaling leading to the activation of p38, p42/44, NF-kB, MyD88 and subsequent generation of tumor necrosis factor alpha and macrophage inflammatory protein 2 [122]. Tumor cell intravasation and metastasis can be promoted by biglycan through the NF-kB and ERK signaling pathways [123].

Another member of the class I SLRP, asporin [111], has been identified as a potential diagnostic marker at the gene level for colorectal [124] and gastric cancers [125,126]. Although asporin is upregulated in pancreatic ductal [127], prostate [128,129] and breast [130] cancers, it acts as a tumor-suppressor gene in triple-negative breast cancer [131]. In scirrhous gastric cancer, asporin is predominantly secreted by CAFs and this leads to the activation of Rac1 through interaction with CD44 [132]. The most significant cancer-related pathways regulated by asporin are FGF2, TGF-β1, BMP-2, epidermal growth factor receptor (EGFR) and CD44 [133].

Aggrecan is deposited in cartilage, the aorta, discs and tendon, and has a role in the viscoelasticity and tensile strength of cartilage together with collagen type II [134]. During the progression of laryngeal squamous cell carcinoma, aggrecan undergoes significant compositional and structural changes, with a stage-related loss of aggregable aggrecan adjacent to apparently normal cartilage and strong presence in advanced stages [135]. Protein levels of this proteoglycan are increased and demonstrate primary localization in tumor epithelial cells in human ovarian cancer tissues [136]. Increased expression levels of aggrecanases (e.g., ADAMTS-1, -4 and -5) and the tissue inhibitor of metalloproteinase 3, which regulates ADAMTS proteolytic activity, have also been seen in ovarian tumors [136]. In contrast, metastatin (also known as a hyaluronan-binding complex), which is released from the aggrecan core protein, can restrict formation of tumor nodules in the lungs of mice injected with B16BL6 melanoma or Lewis lung carcinoma cells [137]. This effect can be explained by the inhibitory effect of metastatin on the migration/proliferation of endothelial cells and VEGF-induced angiogenesis [137]. It is possible that the HA binding motif, present in metastatin, is responsible for this anti-tumorigenic result [138].

Epiphycan belongs to the class III SLRPs, with a known role in cartilage development and joint integrity maintenance [139]. This proteoglycan promotes formation of collagen type I fibers [140]; there is limited information on its role in tumors. Research has shown that epiphycan is differentially expressed in invasive stroma, when compared to the in situ stroma of breast cancer [141]. Moreover, epiphycan has stronger expression in metastatic ovarian cancer than in primary cancer and normal ovaries [142]. The growth and invasion ability of ovarian cancer cells (i.e., SKOV3) can be impaired by *EPYC-siRNA* [142]. Two proteins, phospholipase Cg2 and phosphatidylinositol 4,5-bisphosphate, have been identified as interacting partners of epiphycan, suggesting a role for this proteoglycan in signal transduction [142].

### 4.3. Gamma-Carboxyglutamic-Acid-Containing Proteins

Although there is a substantial amount of evidence suggesting that vitamin K has anticancer potential [143], matrix vitamin-K-dependent proteins have a pro-tumorigenic role in cancer. Six out of fourteen human vitamin-K-dependent proteins play an essential role in skeletal biology [144]. For instance, osteocalcin (also known as bone gamma-carboxyglutamic-acid-containing protein) is the most abundant non-collagenous protein in bone that has upregulated expression in several solid tumors, including ovarian, lung, brain and prostate cancers [145]. Uncarboxylated osteocalcin promotes the development of multiple cancers and has been suggested as a target for preventing bone metastasis in triple-negative breast cancer [146]. Uncarboxylated osteocalcin enhances the proliferation of MDA-MB-231 cells through the TGF-β1/SMAD3 signaling pathway and increases the metastatic potential of cancer cells via upregulation of MMP-2, MMP-13 and VEGF [146]. Although there is no significant difference in the median *BGLAP* mRNA and protein levels between normal pancreas, chronic pancreatitis and ductal adenocarcinoma, osteocalcin is strongly expressed in the cytoplasm and nuclei of tubular complexes and pancreatic intraepithelial neoplasia lesions of diseased human tissues [147].

Mammalian Gla-rich protein (also known as unique cartilage matrix-associated protein and upper zone of growth plate and cartilage matrix-associated protein) is expressed in skeletal and connective tissues (e.g., bone, cartilage, skin and vasculature), and is associated with soft tissue calcification pathologies [148,149]. Mineral formation might be directly influenced by Gla-rich protein [148], as this protein has a high-density γ-carboxyglutamic acid (Gla) domain (higher than in osteocalcin and matrix Gla protein) and, therefore, has stronger binding capacity to calcium ions. Gla-rich protein binds calcium and plays a role in modulating calcium availability in the ECM, acting as a negative regulator of osteogenic differentiation [148,150]. Interestingly, microcalcifications found in breast cancer share similar molecular mechanisms with arterial pathological mineralization, as well as physiological mineralization in bone [151]. In human skin and breast cancers, there is a specific accumulation pattern of carboxylated Gla-rich protein and undercarboxylated GRP; carboxylated Gla-rich protein is lower in non-cancer cells, while uncarboxylated Gla-rich protein has a stronger association with cancer cells [152]. Sites with microcalcifications display high amounts of uncarboxylated Gla-rich protein in breast cancer and invasive ductal carcinoma [152]. Overexpression of Gla-rich protein in triple-negative breast cancer inhibits cancer cell migration, invasion and colony formation (MDA-MB-231 and 4T1 cells, compared to mock control) as well as decreasing tumor growth in in vivo xenograft models [153].

Matrix Gla protein (five-to-six γ-carboxyglutamic acid residues) is secreted by chondrocytes and vascular smooth muscle cells and has an abnormal expression in various cancer types [154]; however, its regulatory role in tumorigenesis is controversial [155]. *MGP* is overexpressed in primary renal cell carcinoma, prostate carcinoma and testicular germ-cell tumors [156], ovarian cancer [157] and precancerous cervical lesions [158]. In gastric cancer, matrix Gla protein interacts with p-STAT5 in the nucleus of cancer cells, where it acts as a transcriptional co-activator through the enhancement of STAT5 binding to target gene promoters [159]. Oncogenic functions of the intracellular matrix Gla protein mainly depend on the JAK2/STAT5 signaling pathway [159]; this protein promotes the proliferation and survival of cancer cells. Matrix Gla protein promotes EMT of triple-negative breast cancer cells and has shown upregulation in clinical breast specimens. These high expression levels are associated with poor, relapse-free survival for triple-negative breast cancer [160] and colorectal cancer patients [161].

### 4.4. Glycoproteins

Thrombospondins are mainly expressed in cartilage, but have also been detected in bone and are produced by bone-resident cells (e.g., osteoblasts, endothelial and immune cells) [162]. Thrombospondin 4 is one of the five members of the thrombospondin family that has low expression in adult tissues, with a dramatic increase during tissue remodeling and regeneration [163]. There is emerging evidence suggesting the involvement of thrombospondin 4 in gastric [164,165], breast [166] and prostate [167] cancers. In gastric tumors, thrombospondin 4 is secreted by CAFs in high quantities, with transcription stimulated by tumor cells [165], and this is associated with cancer metastasis [164]. The invasion of cancer cells can be facilitated by activated stromal response, as has been shown in breast tumors where thrombospondin 4 expression was significantly upregulated in both invasive ductal carcinoma and invasive lobular carcinoma compared to normal stroma [166]. The expression of thrombospondin 4 at high levels promotes cell proliferation and restricts the apoptosis of prostate cancer stem cells by activating the PI3K/Akt pathway; this is coupled with an increase in tumorigenicity in vivo [167]. In contrast, *THBS4* mRNA expression is lower in colorectal tumors compared to matched normal tissues, with the protein typically absent from normal epithelial and cancer cells [168].

Cartilage oligomeric matrix protein (COMP) is almost exclusively expressed in the cartilage and has its main function in ECM assembly by interacting with collagen types I, II and IX, matrilin 3 and fibronectin to promote collagen fibrillogenesis [169,170,171]. However, deposition of this protein in tumor stroma has been observed in breast cancer with a more invasive phenotype, partly due to upregulation of MMP-9 and genes protecting cancer cells from endoplasmic reticulum stress [172]. Remarkably, three peptides derived from COMP do not have a role in the regulation of TGF-β1 signaling and angiogenesis [173]. In periampullary adenocarcinoma, cancer cells and the surrounding stroma exhibit high COMP levels. These expression levels are strongly associated with more aggressive pancreatobiliary-type (PB-type) exclusion of cytotoxic T-cells from the tumors and the presence of a denser collagen matrix [174]. Likewise, COMP is upregulated in bladder and colorectal cancers and is indicative of worse prognosis for the cancer patient [175,176]. Although it has been shown that COMP promotes cancer-cell proliferation via activation of the PI3K/Akt/mTOR/p70S6K signaling pathway [176], more studies are required to decipher the role of this protein in cancer progression and metastasis.

### 4.5. Small Integrin-Binding Ligand N-Linked Glycoproteins

Both bone sialoprotein (also known as integrin-binding sialoprotein) and osteopontin (also known as secreted phosphoprotein 1) are bone matrix proteins, which are produced by osteoblasts, osteocytes, hypertrophic chondrocytes and osteoclasts during bone morphogenesis [177]. These proteins are trapped within mineralized matrices of bones and dentin [178] and have been shown to be significantly upregulated in several epithelial tumors [179], especially those with pathological microcalcifications and tendencies to metastasize to bones. In normal tissues, SIBLINGs function as signal transducers, promoting cell adhesion, motility and survival by binding to a variety of integrins and CD44, and regulators of transcription through NF-κB, regulating cell proliferation and differentiation [180]. Survival of bone-residing metastatic cells (MDA-MB-231) can be promoted via ligation of integrin α_v_β_3_ to sialoprotein-enriched mineralized bone [181,182]; the proliferation of breast cancer cells is also regulated by bone sialoprotein [183]. The growth and aggressiveness of colorectal cancer can be mediated by bone sialoprotein via activation of the Fyn/β-catenin signaling pathway [184]. In non-small-cell lung cancer, levels of bone sialoprotein strongly correlate with bone dissemination and worse overall survival [185]. Interestingly, expression levels of bone sialoprotein correlated with the level of osteopontin in papillary thyroid carcinoma [186]. Indeed, expression of osteopontin in cancer cells contributes to their invasive potential [187,188,189,190,191]. Osteopontin is a major non-collagenous matrix protein, secreted by osteoblasts, osteocytes and other hematopoietic cells, which is involved in osteoclast attachment to mineralized bone matrix and has multiple functions maintaining cell homeostasis. In bladder cancer tissues, *SPP1* is highly expressed and functions of this protein can be mediated by activating the JAK1/STAT1 signaling pathway [192]. Hepatocellular carcinoma uses osteopontin to generate reactive oxygen species for cancer progression. This is facilitated via induction of the JAK2/STAT3/NOX1 signaling pathway [193]. Osteopontin is a ligand for the integrin α_v_β_3_, and, together with activated RANKL, via Rho GTPase, CD44 and MMP-9, plays an important role in prostate cancer cell migration [194]. The osteopontin/ α_v_β_3_ signaling pathway, with the involvement of ERK1/2, regulates expression of VEGF and, therefore, facilitates angiogenesis and prostate cancer progression [195]. Plasma osteopontin can be used like a prostate-specific antigen to predict treatment response in castrate-resistant, metastatic prostate cancer patients after chemotherapy [196].

Dentine matrix protein 1 (DMP-1) is highly expressed by early and mature osteocytes, acting as a transcription factor that nucleates apatite and regulates osteoid mineralization. Data obtained have shown that DMP-1 is expressed in breast and lung cancers and has significant inverse associations with tumor grades [197,198]. Remarkably, the presence of high levels of DMP-1 in primary breast lesions lowered the risk of developing bone metastasis and reduced the migratory capacity of breast cancer cell lines [198]. DMP-1 can induce expression of VE-cadherin, followed by inhibition of VEGFR-2 phosphorylation and Src-mediated signaling, and, therefore, reduce tumor-associated angiogenesis [199]. However, in colon adenocarcinoma, DMP-1 binds and activates proMMP-9 and bridges MMP-9 to CD44, α_v_β_3_ and α_v_β_5_ integrins on the cell surface, enhancing cancer cell invasion and metastasis [200]. Another member of the SIBLING family, dentin sialophosphoprotein (DSPP; isolated from dentine ECM and with low levels in bones), has been shown to be upregulated in human oral squamous cell carcinoma [201]. Silencing of *DSPP* reduced the viability and migration of OSC2 cancer cells and resulted in downregulation of MMP-2, MMP-3, MMP-9, VEGF, Ki-67, p53 and EGFR [201]. Moreover, the loss of *DSPP* significantly downregulated regulators of endoplasmic reticulum stress, including GRP78, SERCA2b, PERK, IRE1 and ATF6, and MMP-20 [202]. DSPP is also ectopically expressed in high-grade prostatic intraepithelial neoplasia and cancerous glands, with pathological stage and Gleason score significantly associated with its expression levels [203]. The expression levels of DSPP have been shown to be lower in normal prostate and thyroid tissues compared to their cancerous counterparts, and this labeling pattern matched MMP-20 expression [204]. DSPP can be processed into dentin sialoprotein (DSP; mainly expressed in odontoblasts), which can bind to integrin β6 and regulate DSPP expression as well as odontoblast homeostasis [205].

Bone metabolism is regulated by matrix extracellular phosphoglycoprotein/osteoblast factor 45 (MEPE/OF45), which is secreted by differentiated osteoblasts and has a marked increase in expression during osteoblast-mediated matrix mineralization. High levels of MEPE in cancer cells correlate with their resistance to ionizing radiation and camptothecin. This is mediated via interaction of MEPE with CHK1 that protects cells from DNA-damage-induced killing [206]. However, the expression profile of this protein across different types of malignancies has not been evaluated in detail to date.

### 4.6. Von Willebrand Factor A Domain-Containing Protein Family

Matrilins are oligomeric extracellular matrix adaptor proteins that mediate interactions between collagen fibrils and other matrix constituents. Collagen fibrillogenesis in cartilage is regulated by matrilin-1 and matrilin-3, which are predominantly expressed in the cartilage [207]; however, their association with cancer remains mainly unclear and requires further study. Matrilin-1 inhibits endothelial cell proliferation and migration by downregulating angiogenesis-related gene markers, including *PECAM1*, *VEGFR* and *VE-cadherin* [208]. Interestingly, *MATN1* mRNA has been shown to be increased in brain metastasis compared to primary breast cancer in which expression levels correlated with patient survival outcomes [209]. Matrilin-1 has limited processing in the cartilage compared to matrilin-3 that is cleaved by ADAMTS-4 and ADAMTS-5 [210]. Upon formation of matrilin-1/-3 hetero-oligomers, neither ADAMTS-5 nor ADAMTS-1 can digest it, probably due to steric hindrance [210]. Matrilin-3 regulates cartilage homeostasis through the induction of *IL-1Ra*, stimulation of *COL2A1* and *ACAN*, and inhibition of *MMP-13* and *ADAMTS-4* and *-5* in chondrocytes, suggesting a chondroprotective role for this protein [211]. The expression level of matrilin-3 is significantly higher in gastric cancer than in normal tissues and is indicative of poor prognosis in patients [212,213].

## 5. Conclusions and Future Directions

Tumors create and develop the surrounding stroma to their specifications. CAFs are the major contributor to these processes via their rich secretome with ECM proteins being an important component. Recognition of the stromal compartment of carcinoma as an important tumor element has long led to the appreciation of multiple opportunities for targeting it as an additional option for cancer treatment. Targeting tumor-specific ECM represents one such opportunity.

The experimental evidence reviewed here unequivocally demonstrates that atypical expression of bone and cartilage ECM proteins is a common phenomenon. Of note, many of these proteins play significant roles in bone and cartilage tissue development, further emphasizing the ability of carcinomas to corrupt and modify tissue-specific developmental programs for tumor advancement. Analyses of cancer gene expression datasets also show that atypical ECM components have significant value as prognostic markers in numerous epithelial tumors (Table 1).

It is evident from the literature that is summarized here that carcinoma-promoting bone and cartilage-specific ECM proteins play very diverse functions in tumor development and progression. Their most prominent homeostatic function, it is logical to suggest, given the major roles of these proteins in tumors, is the regulation of the mechanical properties of tumor tissue, such as its stiffness and viscoelasticity. Alterations of these properties affect a multitude of critical cellular and biochemical processes within the tumor. It is important to emphasize though that the functions of the bone and cartilage ECM proteins are likely to encompass a broad range of signal transducing, mitogenic, metabolic and other roles in carcinoma pathobiology, which are yet to be discovered.

Numerous strategies to target the ECM compartment have been suggested and explored (for recent reviews please see [230,231] and references therein). They include, for example, eliminating cells such as CAFs that produce ECM proteins. This approach has had very limited success so far, with the main reason most likely being the extremely high heterogeneity of the CAF population, where specific, tumor-promoting CAF subsets have only begun to be elucidated. Another popular approach has been to target ECM-modifying enzymes to alter the stroma structure and mechanical properties. Despite holding much promise as anti-cancer drugs, agents that target ECM-modifying enzymes have also proved to be mostly disappointing since the usefulness of non-selective MMP and LOXL inhibitors is limited by their significant toxicity and failure to show any objective clinical response. Therefore, considering the above, targeting tumor stroma-specific ECM with no known homeostatic function in the tumor tissue of origin, such as bone and cartilage proteins in carcinoma, seems like an attractive alternative strategy for future anti-cancer therapies. Atypical ECM proteins may also represent attractive components for designing future drug-delivery systems due to their limited tissue distribution and restricted homeostatic function.

From the published data reviewed here, it is clear that multiple bone and cartilage-specific ECM components play significant roles in numerous types of carcinomas and, thus, may represent attractive targets for novel stroma-targeting therapies. However, numerous outstanding questions will need to be answered before designing such drugs will become possible. Specifically, it is important to identify cellular populations that produce these atypical ECM components and to discover what specific signals within the tumor stimulate their production. Applying emerging proteomics and multiplex spatial-imaging techniques will allow qualitative and quantitative assessment of the intra-tumoral distribution of bone and cartilage ECM proteins, which is currently unavailable, but may lead to the identification of novel ECM-based biomarkers for personalized cancer treatments. Additionally, characterizing precise functions of these “unusual suspects” in the tumor microenvironment will aid the development of more efficient and selective stroma-targeting therapies for carcinomas and, potentially, other malignancies. 

## Figures and Tables

**Figure 1 cancers-15-00791-f001:**
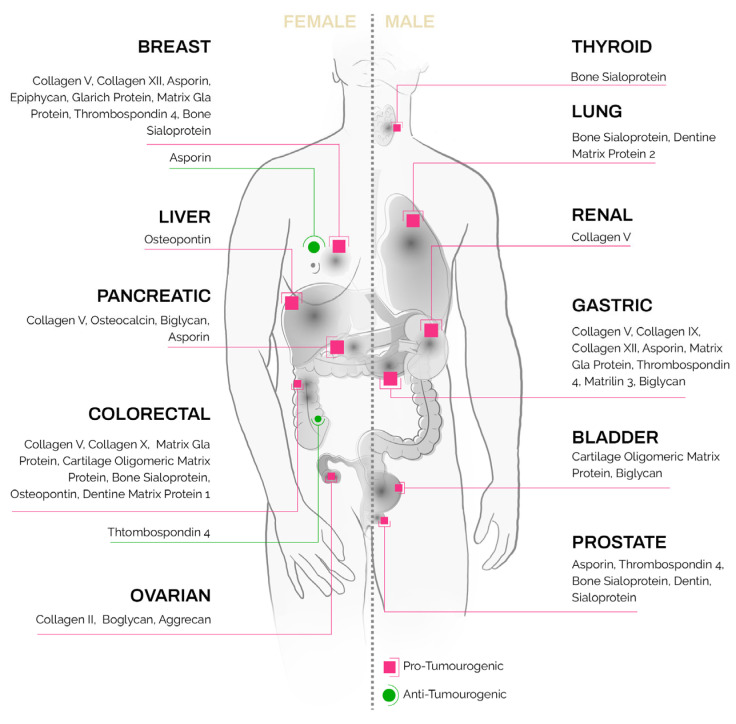
Bone- and cartilage-specific ECM proteins found in carcinoma subtypes.

**Table 1 cancers-15-00791-t001:** Expression of bone- and cartilage-specific matrix proteins in epithelial cancers.

ECM Component	Gene	Protein Expression	Carcinoma	Prognostic Marker
**Collagenous Proteins**				
**Collagen II**	*COL2A1*	Specific for cartilaginous tissues and the vitreous humor of the eye.	Pro-tumorigenic in high-grade serous carcinoma [74]	
**Collagen V**	*COL5A1, COL5A2, COL5A3*	Minor connective tissue component of nearly ubiquitous distribution, found in tissues containing type I collagen.	Pro-tumorigenic in breast invasive ductal carcinoma [77]; colorectal cancer [80]; gastric cancer, including renal metastasis [214,215]; pancreatic ductal adenocarcinoma, with promotion of hepatic metastasis [84]; co-expression of high (COL5A1)2/low (COL5A2) heterotrimer is unfavorable in tongue squamous cell carcinoma [81].	Prognostic marker in colorectal cancer (unfavorable [80]), renal cancer (unfavorable), glioma (unfavorable), urothelial cancer (unfavorable) and lung cancer (unfavorable).
**Collagen IX**	*COL9A1, COL9A2, COL9A3*	Expressed in hyaline cartilage and vitreous of the eye.	Pro-tumorigenic in gastric cancer [88].	Prognostic marker in endometrial cancer (unfavorable).
**Collagen X**	*COL10A1*	Hypertrophic chondrocytes and presumptive mineralization zones of hyaline cartilage.	Pro-tumorigenic in breast cancer [93]; colorectal cancer (in conjugation with visinin-like 1 protein [98]).	
**Collagen XII**	*COL12A1*	Found in association with type I collagen.	Pro-tumorigenic in breast cancer [102]; gastric cancer [103].	Prognostic marker in renal cancer (unfavorable).
**Collagen XXIV**	*COL24A1*	Expressed in differentiating osteoblasts [104] and developing cornea [216].	Pro-tumorigenic in squamous cell carcinoma of the head and neck (*COL24A1* mRNA [106]); hepatocellular carcinoma (*COL24A1* mRNA [107]).	
**Proteoglycans**				
**Biglycan**	*BGN*	Expressed mainly in tendon, cartilage and bone.	Pro-tumorigenic in endometrial cancer [118]; esophageal squamous cell carcinoma [217]; gastric cancer [218]; head and neck squamous cell cancer [219]; pancreatic cancer [113]; urothelial carcinoma of bladder [220].	Prognostic marker in colorectal cancer [221]; gastric cancer [222]; renal cancer (unfavorable).
**Asporin**	*ASPN*	Expressed by osteoblasts.	Pro-tumorigenic in breast cancer [130]; gastric cancer [132]; pancreatic cancer [127]; prostate cancer [128,129]. Anti-tumorigenic in breast cancer [131].	Prognostic marker in renal cancer (unfavorable).
**Aggrecan**	*ACAN*	Major component of extracellular matrix of cartilagenous tissues.	Pro-tumorigenic in ovarian cancer [136].	Prognostic marker in renal cancer (unfavorable).
**Epiphycan**	*EPYC*	Expressed in cartilage and testis [223].	Pro-tumorigenic in ovarian cancer, with higher expression in metastatic cancer than in primary cancer [142].	Prognostic marker in pancreatic cancer (unfavorable).
**Gamma-carboxyglutamic Acid-containing Proteins**				
**Osteocalcin**	*BGLAP*	Highly abundant bone protein secreted by osteoblasts. Constitutes ~1–2% of the total bone protein.	Pro-tumorigenic in breast cancer [146]; pancreatic cancer [147].	
**Gla-rich protein**	*UCMA*	Expressed in the upper immature zone of fetal and juvenile epiphyseal cartilage.	Both γ-carboxylated GRP (cGRP)/undercarboxylated GRP (ucGRP) are found in healthy tissues, while ucGRP is the predominant form associated with tumor cells in skin and breast cancer [152].	
**Matrix Gla protein**	*MGP*	Expressed by chondrocytes and vascular smooth muscle cells. Associates with the organic matrix of bone and cartilage.	Pro-tumorigenic in breast cancer [154,160]; colorectal cancer [161]; gastric cancer [159].	Prognostic marker in renal cancer (unfavorable).
**Glycoproteins**				
**Thrombospondin 4**	*THBS4*	Expressed in the articular cartilage, also been detected in bone, restricted to the osteoblast lineage [224,225].	Pro-tumorigenic in breast cancer [166]; gastric cancer [164,165]; high level of expression in stem cells in prostate cancer [167]. Anti-tumorigenic in colorectal cancer [168].	Prognostic marker in urothelial cancer (unfavorable).
**Cartilage oligomeric matrix protein**	*COMP*	Expressed by osteoblasts in bone and cartilage during embryogenesis [226].	Pro-tumorigenic in periampullary adenocarcinoma [174]; bladder cancer [175]; colorectal cancer [176].	Prognostic marker in renal cancer (unfavorable), colorectal cancer (unfavorable), endometrial cancer (unfavorable) and urothelial cancer (unfavorable).
**Small Integrin-binding Ligand N-linked Glycoproteins**				
**Bone sialoprotein**	*IBSP*	Major structural protein of the bone matrix (~12% non-collagenous proteins). Expressed in hypertrophic chondrocytes, osteoblasts, osteocytes, osteoclasts and trophoblasts.	Pro-tumorigenic in breast and prostate cancers [227]; colorectal cancer [184]; non-small-cell lung cancer [185]; thyroid cancer [186].	
**Osteopontin**	*SPP1*	Expressed by osteoblasts, odontoblasts and osteocytes.	Pro-tumorigenic in colorectal cancer [187]; head and neck carcinoma [188]; hepato-cellular carcinoma [193].	Prognostic marker in liver cancer (unfavorable), pancreatic cancer (unfavorable) and cervical cancer (unfavorable).
**Dentine matrix protein 1**	*DMP1*	Expressed in odontoblasts, ameloblasts and cementoblasts, as well as in fully differentiated osteoblasts in bones.	Pro-tumorigenic in colorectal cancer [200]; lung cancer [197]. Altered DMP1 splicing in breast cancer [228].	
**Dentin sialophosphoprotein**	*DSPP*	Expressed by odontoblasts and is proteolytically processed to generate dentin sialoprotein and dentin phosphoprotein.	Pro-tumorigenic in prostate cancer [203]; oral squamous cell carcinoma [229].	
**Matrix extracellular phosphoglycoprotein**	*MEPE*	Expressed in odontoblasts, osteoblasts, and osteocytes.	High levels of MEPE in cancer cells correlate with their resistance to ionizing radiation and camptothecin [206].	
**von Willebrand Factor A Domain-containing Proteins**				
**Matrilin 1**	*MATN1*	Major component of the extracellular matrix of non-articular cartilage.	Pro-tumorigenic in metastatic breast cancer (*MATN1* mRNA [209]).	
**Matrilin 3**	*MATN3*	Major component of the extracellular matrix of cartilage.	Pro-tumorigenic in gastric cancer [212].	Prognostic marker in stomach cancer (unfavorable), liver cancer (unfavorable) and cervical cancer (unfavorable).

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
