# Peer review of "Unusual Suspects: Bone and Cartilage ECM Proteins as Carcinoma Facilitators"

_cancers, 2023, doi:10.3390/cancers15030791_

Round 1

Reviewer 1 Report

This is a very nicely written review on the role of bone and cartilage ECM proteins in development of carcinoma. I have enjoyed reading this excellent review. The authors should be praised for high-quality literature review on the topic. I do not have any major comments. However, I have several minor comments:

1) Conclusions are too vague and full of generic phrases. Please provide more constructive conclusions. Please bring more synthesis.

2) I would expect more meaningful figures in this review. Currently, the Figure 1 is the only figure and it is not very insightful. 

Author Response

Review Report #1

Comments and Suggestions for Authors:

This is a very nicely written review on the role of bone and cartilage ECM proteins in development of carcinoma. I have enjoyed reading this excellent review. The authors should be praised for high-quality literature review on the topic. I do not have any major comments. However, I have several minor comments:

1) Conclusions are too vague and full of generic phrases. Please provide more constructive conclusions. Please bring more synthesis.

This section has been modified according to reviewer’s instructions.

2) I would expect more meaningful figures in this review. Currently, the Figure 1 is the only figure, and it is not very insightful.

We thank the Reviewer for pointing this out. The Figure 1 is a schematic illustration of the outlined information represented in Table 1. This review summarizes and critically assesses available information on the expression and function of the atypical for the carcinoma bone- and cartilage specific components of the extracellular matrix proteins (Table 1).

Reviewer 2 Report

The review of Sorvina et al. provides an overview of the literature data on the role of bone and cartilage ECM proteins in the pathogenesis of carcinoma. The Authors describe cancer-related functions of the ECM, bone- and cartilage-specific matrisome, and the role of ECM components in cancer progression, spread and invasion. The manuscript has a structure and merit, and is clearly written. For the Authors’ consideration, I would suggest several modifications that would definitely improve the impact of the paper:

-          the rational for focusing on bone/cartilage ECM only should be more explicitly provided, as it might not be obvious for the broad audience;

-          in several cases the Authors shortly refer to ECM effects on cell physiology. In this context and in the context of the story topic, the mechanistic description of the impact of ECM on cell proliferation, differentiation, apoptosis, would add to the message of the story;

-          in particular, the differences in reactivity of cancer and normal cells to physical interactions with the niches, the function of integrin/cadherin-mediated cell/cell/ECM contacts  in the regulation GF-mediated signaling/mechanotransduction, etc., could be discussed;

-          then, the discussion of the prospective use of  the ECM components characteristic for bone and cartilage as  drug targets is superficial. The Authors could be more specific at pointing out the possible tools for targeting  ECM specifically in the tumors.

Author Response

Review Report #2

Comments and Suggestions for Authors:

The review of Sorvina et al. provides an overview of the literature data on the role of bone and cartilage ECM proteins in the pathogenesis of carcinoma. The Authors describe cancer-related functions of the ECM, bone- and cartilage-specific matrisome, and the role of ECM components in cancer progression, spread and invasion. The manuscript has a structure and merit, and is clearly written. For the Authors’ consideration, I would suggest several modifications that would definitely improve the impact of the paper:

  • the rational for focusing on bone/cartilage ECM only should be more explicitly provided, as it might not be obvious for the broad audience;

We thank the Reviewer for their useful suggestion. The following clarification has been added (Page 2, lines 65-68):

“Cancer that arises in the epithelia is the most common type and is found in most organs of the human body. In this Review, we summarize cancer-specific expression patterns of bone and cartilage ECM proteins and their roles in specific tumor types; as an overview of these ectopically expressed ECM proteins in cancer is missing.”

  • in several cases the Authors shortly refer to ECM effects on cell physiology. In this context and in the context of the story topic, the mechanistic description of the impact of ECM on cell proliferation, differentiation, apoptosis, would add to the message of the story;

We would like to thank the Reviewer for their t insightful comment. The following text has been added:

(Page 3, lines 100-112)

“Perturbation of the ECM homeostasis affects biochemical and biophysical properties of the matrix, both crucial for the normal tissue function. For example, in breast cancer…”

(Page 3, lines 123-125)

“Transmission of the mechanical stress (e. g. osmotic compression) to the cells is mediated by the ECM and this impacts cell proliferation and migration in 3D environment [33704063].”

Detailed description on the role of the ECM on cell differentiation has been presented on page 2, lines 83-94:

“The potent stimulator of fibrosis is transforming growth factor beta (TGF-β1) that can en-hance expression of ECM remodeling gene ACTA2 (alpha-smooth muscle actin (α-SMA)) [11], PLOD2, encoding procollagen-lysine,2-oxoglutarate 5-dioxygenase 2 that is required for the collagen deposition [12,13] and numerous other pro-fibrotic proteins…”

(Page 2, lines 51-53)

“A large body of experimental and clinical evidence have shown that altered expression of matrices in tumors associates with cancer onset and progression, and response to the treatment due to the altered apoptosis [36419159].”

  • in particular, the differences in reactivity of cancer and normal cells to physical interactions with the niches, the function of integrin/cadherin-mediated cell/cell/ECM contacts in the regulation GF-mediated signaling/mechanotransduction, etc., could be discussed;

The main idea behind this literature review was to attract attention of the tumour-stroma - research community to the findings of altered tissue specificity, and carcinoma-related functions of the ECM components that are mainly found in bone and cartilage with the view to provide different perspective on the regulation of the ECM compartment in epithelial tumours. Unfortunately, there is still insufficient information available on the specific aspects of the pathobiology of this “subclass” of the ECM proteins to cover the subjects that are  suggested by the reviewer.

  • then, the discussion of the prospective use of the ECM components characteristic for bone and cartilage as drug targets is superficial. The Authors could be more specific at pointing out the possible tools for targeting ECM specifically in the tumors.

This section has been modified according to reviewer’s instructions.